# Oxic-anoxic regime shifts mediated by feedbacks between biogeochemical processes and microbial community dynamics

Timothy Bush[1,2], Muhe Diao[1], Rosalind J. Allen[2], Ruben Sinnige[1], Gerard Muyzer[1] & Jef Huisman[1]

Although regime shifts are known from various ecosystems, the involvement of microbial communities is poorly understood. Here we show that gradual environmental changes induced by, for example, eutrophication or global warming can induce major oxic-anoxic regime shifts. We first investigate a mathematical model describing interactions between microbial communities and biogeochemical oxidation-reduction reactions. In response to gradual changes in oxygen influx, this model abruptly transitions between an oxic state dominated by cyanobacteria and an anoxic state with sulfate-reducing bacteria and photo-trophic sulfur bacteria. The model predictions are consistent with observations from a seasonally stratified lake, which shows hysteresis in the transition between oxic and anoxic states with similar changes in microbial community composition. Our results suggest that hysteresis loops and tipping points are a common feature of oxic-anoxic transitions, causing rapid drops in oxygen levels that are not easily reversed, at scales ranging from small ponds to global oceanic anoxic events.

[1] Department of Freshwater and Marine Ecology, Institute for Biodiversity and Ecosystem Dynamics, University of Amsterdam, P.O. Box 94248, Amsterdam 1090 GE, The Netherlands. [2] SUPA, School of Physics and Astronomy, University of Edinburgh, Edinburgh EN9 3FD, UK. Correspondence and requests for materials should be addressed to G.M. (email: g.muijzer@uva.nl)

Ecosystems may undergo sharp transitions in response to smooth environmental changes[1–4]. If these transitions lead to large and persistent changes in ecosystem structure and functioning, they are generally referred to as regime shifts. Regime shifts are often attributed to alternative stable states (where distinct ecological states exist under the same external conditions[1, 2]) and have been documented in a wide range of terrestrial, freshwater, and marine habitats[3, 4]. Theoretical work has contributed to understanding ecological regime shifts[3] and has identified early warning signs that a regime shift may be imminent[4]. However, so far the potential involvement of microbial communities in regime shifts has been largely ignored, despite the fact that microbial communities make a significant contribution to many ecological and biogeochemical processes[5, 6].

A small number of studies suggest that regime shifts may occur in microbial communities. Recent work has pointed at the existence of alternative stable states in the microbial community of the human gut[7, 8] and in phytoplankton populations sensitive to high light[9, 10]. Other studies have shown a regime shift from iron reduction to sulfate reduction in an iron-rich groundwater flow[11] and compositional regime shifts in a nitrifying batch reactor[12]. However these examples are few, and furthermore, unlike in traditional macro-ecology, there is little theoretical work on regime shifts in microbial ecosystems[13, 14].

Microorganisms play an important role in the biogeochemical cycles of lakes and oceans. Many waters in temperate climates become vertically stratified in summer, when the sun heats up the surface layer while the temperature in the deeper layers remains low. Seasonal stratification induces changes in microbial community structure[15–17]. The surface layer (epilimnion) is usually rich in oxygen and often dominated by oxygenic phototrophic microorganisms such as cyanobacteria and eukaryotic algae. Especially in productive waters, microbial degradation of organic matter can create anoxic conditions in the deeper layers (hypolimnion), which may shift the microbial community to heterotrophic bacteria utilizing nitrate or sulfate as an electron acceptor. In between these layers, in the metalimnion, oxygen diffusing down from the epilimnion meets sulfide diffusing up from the hypolimnion, providing a niche for photosynthetic and non-photosynthetic sulfur-oxidizing bacteria[18, 19]. Because of surface cooling and wind action, the stratification is broken in the fall when different water layers are mixed, homogenizing the oxygen and temperature profiles.

In this paper, we combine a mathematical model and lake data, to show that regime shifts in microbial community structure may be common in seasonally stratified waters with an active microbial sulfur cycle. We first present a simple mathematical model of a microbial ecosystem containing cyanobacteria, sulfate-reducing bacteria and phototrophic sulfur bacteria. We show that this model can undergo regime shifts between oxic and anoxic states in response to gradual parameter variations that mimic changes in vertical stratification and hence oxygen diffusivity across the thermocline. Subsequently, we compare the model predictions with data from a seasonally stratified lake with an anoxic hypolimnion during summer, and discuss the wider implications of oxic-anoxic regime shifts for other ecosystems.

## Results

**Bringing microbial dynamics into a biogeochemical model.** We consider a simple model that integrates microbial community dynamics with biogeochemical processes. The microbial community consists of three functional groups: oxygen-producing cyanobacteria (CB), phototrophic sulfur bacteria (PB) such as purple or green sulfur bacteria, and sulfate-reducing bacteria (SB) (Fig. 1). Growth of each microbial population is

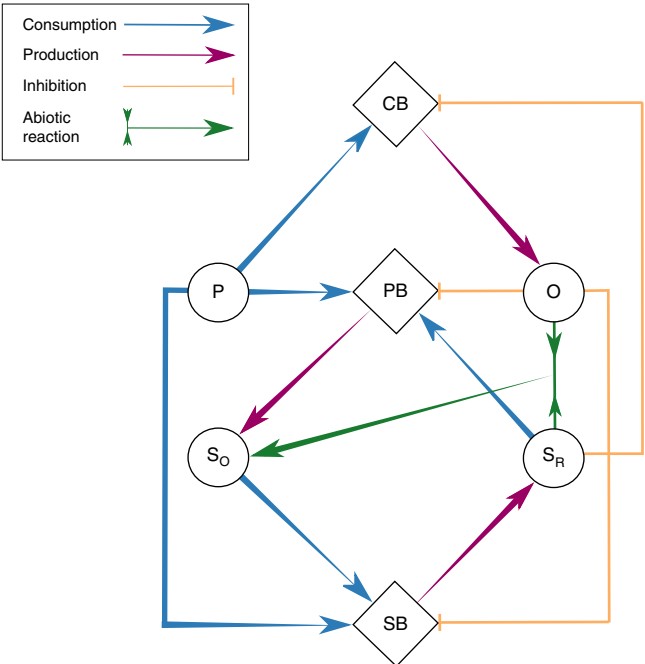

**Fig. 1** Schematic diagram of the microbial ecosystem model. The model consists of three bacterial functional groups (CB cyanobacteria, PB phototrophic sulfur bacteria, SB sulfate-reducing bacteria) and four chemical substrates (P phosphorus, O oxygen, $S_R$ reduced sulfur, $S_O$ oxidized sulfur). *Arrows* denote the consumption (*blue arrows*) and production (*magenta arrows*) of chemicals by the microbial populations. *Orange lines* represent growth inhibition of the microbial populations. *Green arrows* indicate abiotic oxidation of reduced sulfur to oxidized sulfur

assumed to depend on the availability of phosphorus (P), a key limiting resource for many aquatic ecosystems[20]. Furthermore, the growth of phototrophic sulfur bacteria depends on sulfide ($S_R$, representing reduced sulfur), whereas that of sulfate-reducing bacteria depends on sulfate ($S_O$, representing oxidized sulfur). Sulfide produced by sulfate-reducing bacteria inhibits the growth of cyanobacteria[21]. Conversely, oxygen (O) produced by cyanobacteria is assumed to be inhibitory to both sulfate-reducing bacteria[22] and phototrophic sulfur bacteria[23].

Hence, changes in the population densities of cyanobacteria ($N_{CB}$), phototrophic sulfur bacteria ($N_{PB}$) and sulfate-reducing bacteria ($N_{SB}$) can be described as:

$$\frac{dN_{CB}}{dt} = g_{CB}(P)\, h_{CB}(S_R)N_{CB} - m_{CB}N_{CB} \tag{1}$$

$$\frac{dN_{PB}}{dt} = g_{PB}(P, S_R)\, h_{PB}(O)N_{PB} - m_{PB}N_{PB} \tag{2}$$

$$\frac{dN_{SB}}{dt} = g_{SB}(P, S_O)h_{SB}(O)N_{SB} - m_{SB}N_{SB} \tag{3}$$

where the functions $g_j(X,Y)$ and $h_j(X)$ describe growth and inhibition, respectively, of microbe $j$ on substrates $X$ and $Y$, and $m_j$ is the mortality rate of microbe $j$ (e.g., due to grazing or viral lysis). The growth and inhibition functions are detailed in the Methods section.

Reduced sulfur is oxidized by phototrophic sulfur bacteria, whereas oxidized sulfur is reduced by sulfate-reducing bacteria, connecting these species in a biogeochemical cycle as they pass the sulfur back and forth (Fig. 1). In addition, oxidation of reduced sulfur can be mediated by chemolithotrophic sulfur-

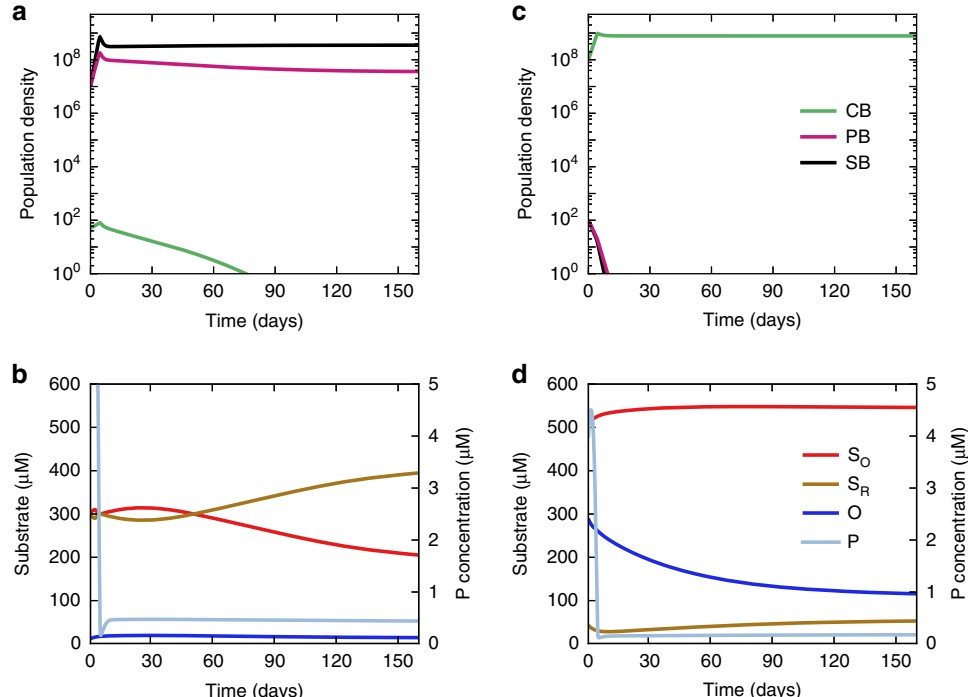

**Fig. 2** Illustration of the two alternative stable states. **a**, **b** When the model starts with a low initial population density of cyanobacteria (CB), it develops towards an anoxic ecosystem with, **a** high abundances of phototrophic sulfur bacteria (PB) and sulfate-reducing bacteria (SB) and, **b** a high concentration of reduced sulfur ($S_R$) but low oxygen concentration (O). **c**, **d** When the model starts with a high initial population density of cyanobacteria, it develops towards an oxic ecosystem with, **c** high abundances of cyanobacteria and, **d** high concentrations of oxygen and oxidized sulfur ($S_O$). Parameter values are given in Supplementary Table 1, with $P_b = 9.5\ \mu M$, $\alpha_O = 8 \times 10^{-4}\ h^{-1}$. Initial conditions: **a**, **b** $N_{PB} = N_{SB} = 1 \times 10^7$ cells $L^{-1}$, $N_{CB} = 5 \times 10^1$ cells $L^{-1}$, $S_O = 300\ \mu M$, $S_R = 300\ \mu M$, $O = 10\ \mu M$, $P = 10\ \mu M$; **c**, **d** $N_{PB} = N_{SB} = 1 \times 10^2$ cells $L^{-1}$, $N_{CB} = 1 \times 10^8$ cells $L^{-1}$, $S_O = 500\ \mu M$, $S_R = 50\ \mu M$, $O = 300\ \mu M$, $P = 4\ \mu M$

oxidizing bacteria and may also occur abiotically[24], which we have modeled here as a simple first-order process. Furthermore, we assume a small diffusive flux of oxidized and reduced sulfur into or out of the system, depending on their concentration gradients and a substrate-specific diffusivity $\alpha_S$. Accordingly, changes in the oxidized and reduced sulfur concentrations can be described as:

$$\frac{dS_O}{dt} = \frac{1}{y_{PB}^S} g_{PB}(P, S_R)\, h_{PB}(O) N_{PB}$$
$$- \frac{1}{y_{SB}^S} g_{SB}(P, S_O)\, h_{SB}(O) N_{SB} \qquad (4)$$
$$+ cOS_R + \alpha_S(S_{O,b} - S_O)$$

$$\frac{dS_R}{dt} = - \frac{1}{y_{PB}^S} g_{PB}(P, S_R)\, h_{PB}(O) N_{PB}$$
$$+ \frac{1}{y_{SB}^S} g_{SB}(P, S_O)\, h_{SB}(O) N_{SB} \qquad (5)$$
$$- cOS_R + \alpha_S(S_{R,b} - S_R)$$

where $y_j^k$ is the yield (in cells per µmole of substrate) of microbe $j$ on substrate $k$, $c$ is the oxidation rate of reduced sulfur, and $S_{O,b}$ and $S_{R,b}$ are the background concentrations of oxidized and reduced sulfur, respectively.

Oxygen is produced by cyanobacteria, reacts with reduced sulfur, and diffuses into or out of the system depending on the concentration gradient. Finally, phosphorus is consumed by all three microbial groups, and also has a diffusive flux across the system boundary. Hence, the oxygen and phosphorus dynamics

can be written as:

$$\frac{dO}{dt} = p_{CB} g_{CB}(P)\, h_{CB}(S_R) N_{CB} - cOS_R + \alpha_O(O_b - O) \qquad (6)$$

$$\frac{dP}{dt} = - \frac{1}{y_{CB}^P} g_{CB}(P) h_{CB}(S_R) N_{CB}$$
$$- \frac{1}{y_{PB}^P} g_{PB}(P, S_R) h_{PB}(O) N_{PB} \qquad (7)$$
$$- \frac{1}{y_{SB}^P} g_{SB}(P, S_O) h_{SB}(O) N_{SB} + \alpha_P(P_b - P)$$

where $p_{CB}$ is the oxygen production per cyanobacterial cell. Parameter values were based on microbial communities of freshwater lakes, where available (Supplementary Table 1). We note, however, that the model results are quite robust, since we obtained qualitatively similar results when using parameter values representative of marine environments[25] or microbial mats[26].

**Oxic-anoxic regime shifts**. Ecological regime shifts can be identified by certain behaviors[3, 13]. First, models with regime shifts often contain alternative stable states, such that they may develop in different directions depending on the initial conditions. Second, environmental changes can cause ecosystems containing alternative stable states to become stuck in an ecosystem state, such that simply reversing the environmental change is not sufficient to return the ecosystem to its original state: a phenomenon known as hysteresis. For example, overfishing can cause collapses in coral reef structure that cannot be recovered simply by returning to earlier, lower fishing rates[27]. Here, we demonstrate that our model displays these two characteristic signs of regime shifts by investigating its response

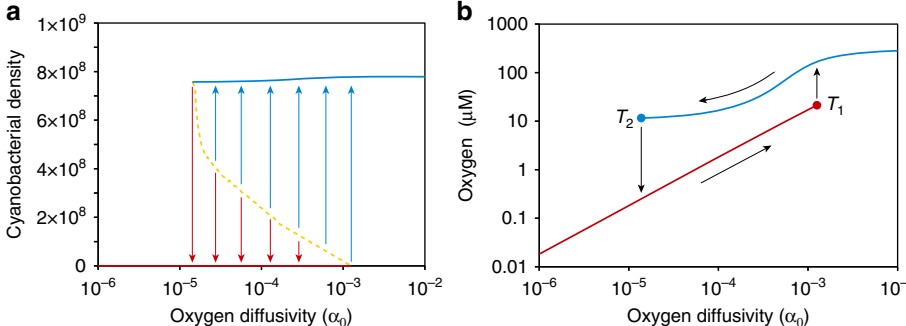

**Fig. 3** Regime shifts between oxic and anoxic states. **a** Cyanobacterial population density and, **b** oxygen concentration predicted at steady state, as function of the oxygen diffusivity. *Blue lines* indicate the oxic state and *red lines* the anoxic state. In **a** the *blue* and *red arrows* indicate the basins of attraction of the oxic and anoxic state, respectively, and the *dashed orange line* is the separatrix between these two basins of attraction. In **b** $T_1$ and $T_2$ indicate the two tipping points of the system and *black arrows* illustrate the hysteresis loop. Parameter values are given in Supplementary Table 1, with $P_b = 9.5\,\mu M$. The initial cyanobacterial density varies, while the other initial conditions are set at $N_{PB} = N_{SB} = 1 \times 10^8$ cells $L^{-1}$, $S_O = 250\,\mu M$, $S_R = 350\,\mu M$, $O = 150\,\mu M$, $P = 9.5\,\mu M$

to parameter changes designed to mimic seasonal variation in lake stratification.

First, we investigate the sensitivity of the model to the initial composition of the microbial community. Figure 2 compares how the ecosystem develops over time for different initial bacterial population densities. The results show that the model is sensitive to the initial community composition, demonstrating the existence of alternative stable states. If sulfate-reducing and phototrophic sulfur bacteria are initially dominant, the oxygen concentration remains low, the concentration of reduced sulfur becomes sufficiently high to suppress cyanobacterial growth, and the system develops an anoxic state (Fig. 2a, b). Conversely, if cyanobacteria are dominant first, their photosynthetic oxygen production is sufficiently high to suppress the growth of sulfate-reducing bacteria and phototrophic sulfur bacteria, generating an oxic state (Fig. 2c, d).

Second, to determine whether the model displays hysteresis, we investigate how it responds to changes in oxygen influx. We vary the parameter $\alpha_O$ as a proxy for the turbulent diffusion of oxygen across the thermocline of a seasonally stratified lake. A low value of the oxygen diffusivity $\alpha_O$ represents the case where stratification restricts turbulent diffusion of oxygen across the thermocline, and a high value of oxygen diffusivity represents the case where the lake is completely mixed and oxygenated. The results confirm the existence of two alternative stable states for a wide range of oxygen diffusivities: one with a stable cyanobacterial population at steady state (blue line) and another where the cyanobacterial population has collapsed (red line) (Fig. 3a). Towards which of these two states the system develops depends on whether the initial population density of cyanobacteria is above or below the separatrix (orange dashed line).

The two alternative stable states are also visible in the steady-state concentration of dissolved oxygen: the ecosystem becomes either oxic or anoxic (Fig. 3b). Starting from the anoxic state, the ecosystem undergoes a regime shift when the oxygen diffusivity increases to high levels. That is, at the tipping point $T_1$ it abruptly transitions from the anoxic to the oxic state. Conversely, once the ecosystem is oxic, it remains oxic when the oxygen diffusivity decreases, and flips back to the anoxic state only when oxygen diffusivity has become very low at tipping point $T_2$. Hence, the system displays hysteresis (as indicated by black arrows in Fig. 3b). The underlying mechanism for the existence of this hysteresis loop is that the microbial community composition differs between the oxic and anoxic states. The anoxic state is characterized by dense populations of phototrophic sulfur bacteria and sulfate-reducing bacteria and high sulfide

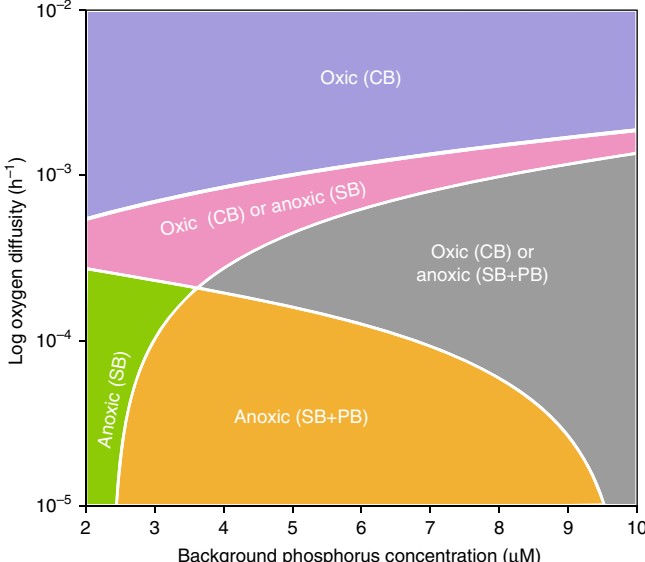

**Fig. 4** Two-parameter bifurcation diagram. The diagram illustrates how the model predictions vary with phosphorus availability and oxygen diffusivity. *Blue region*=oxic state with cyanobacteria. *Pink region*=alternative stable states, with either cyanobacteria in the oxic state or sulfate-reducing bacteria in the anoxic state. *Gray region* = alternative stable states, with either cyanobacteria in the oxic state or coexistence of sulfate-reducing bacteria and phototrophic sulfur bacteria in the anoxic state. *Green region*=anoxic state with sulfate-reducing bacteria. *Orange region*=anoxic state with coexistence of sulfate-reducing bacteria and phototrophic sulfur bacteria. Parameter values are given in Supplementary Table 1

concentrations, which prevent cyanobacterial invasion over a wide range of oxygen diffusivities. Only when the oxygen influx becomes high enough to suppress the phototrophic sulfur and sulfate-reducing bacteria, can the cyanobacteria invade. Conversely, the oxic state is dominated by cyanobacteria whose photosynthetic oxygen production contributes to the persistence of the oxic state, thereby suppressing invasion of the anaerobic sulfur bacteria. Thus, only at a very low oxygen diffusivity can the anaerobic sulfur bacteria become established.

The anoxic state of our model can take two different forms depending on the oxygen input. At low oxygen diffusivity, the model predicts coexistence of sulfate-reducing bacteria and

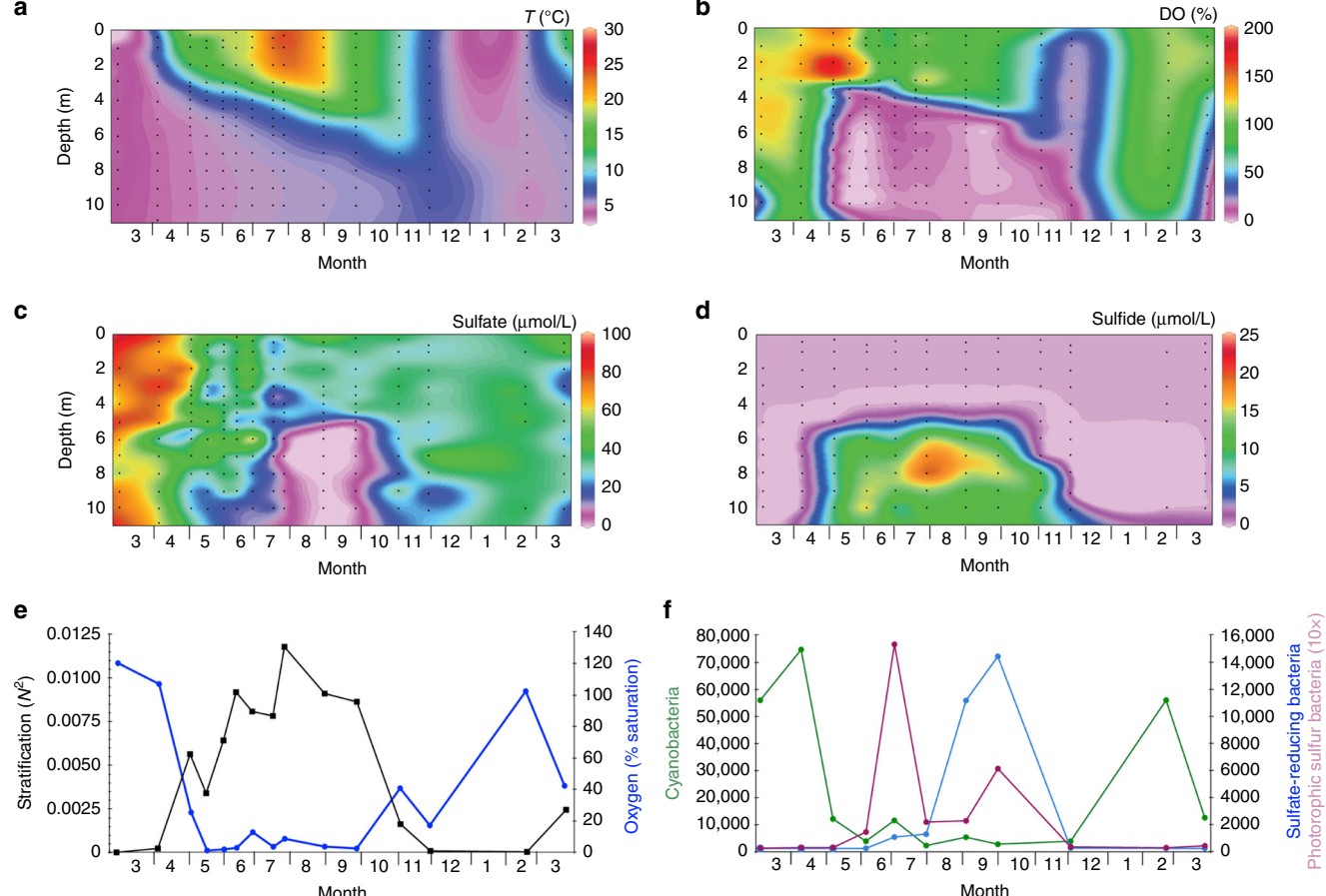

**Fig. 5** Data from the seasonally stratified Lake Vechten. Contour plots of seasonal changes in, **a** temperature; **b** dissolved oxygen; **c** sulfate; **d** sulfide. The plots are based on 189 water samples (*black dots*) taken from March 2013 to early April 2014. **e** Seasonal changes of the strength of stratification (*black line*, quantified by the squared buoyancy frequency $N^2$ at the thermocline) and oxygen saturation just below the thermocline (*blue line*, measured at 6 m depth). **f** Seasonal changes in relative abundances of cyanobacteria (*green line*), phototrophic sulfur bacteria (*magenta line*) and sulfate-reducing bacteria (*light blue line*) in the metalimnion (at 6 m depth), based on 16S rRNA sequence data

phototrophic sulfur bacteria which pass the sulfur back and forth (orange and gray areas in Fig. 4). At intermediate oxygen diffusivity, the anoxic state consists of sulfate-reducing bacteria only (green and pink area). In this case, oxidation of reduced sulfur by sulfur-oxidizing bacteria and direct chemical oxidation diverts reduced sulfur from the phototrophic sulfur bacteria, while resupplying sulfate-reducing bacteria with oxidized sulfur. Hence, sulfate-reducing bacteria obtain a selective advantage, and can displace the phototrophic sulfur bacteria during competition for phosphorus.

Phosphorus enrichment has a profound effect on oxic-anoxic regime shifts. As phosphorus availability increases, the model predicts that the region with alternative stable states spreads out over a larger range of oxygen diffusivities (gray and pink area in Fig. 4). The reason is that, in the oxic state, the population density of cyanobacteria increases with phosphorus enrichment. High cyanobacterial densities can sustain the oxic state by their own photosynthetic oxygen production even when the diffusive oxygen influx into the system becomes very low. Conversely, in the anoxic state, population densities of the anaerobic sulfur bacteria increase with phosphorus enrichment, and hence they can maintain anoxic conditions up to higher oxygen diffusivities. Thus, the region with alternative stable states widens with phosphorus enrichment, suggesting that particularly eutrophic waters will be very sensitive to oxic-anoxic regime shifts.

**Comparison with lake data**. Our model is a simplification of reality, in which we have reduced the highly diverse microbial communities and plethora of biogeochemical reactions in aquatic ecosystems to only a few interacting processes (Fig. 1). It is therefore interesting to assess whether we can find signatures for oxic-anoxic regime shifts in lakes that are consistent with the model results.

We investigated microbial community dynamics and water quality parameters in the seasonally stratified Lake Vechten[18, 28], a small lake in the Netherlands. In early spring, the lake was well mixed and oxygenated (Fig. 5a, b). Subsequently, a temperature stratification developed, with warm oxygen-saturated waters in the epilimnion whereas the hypolimnion became anoxic during summer. Nitrate concentrations (not shown) were < 1 μM in the hypolimnion, providing favorable conditions for sulfate reduction. Sulfate concentrations were highest in early spring, and decreased to low concentrations in the anoxic hypolimnion (Fig. 5c). Conversely, sulfide accumulated in the anoxic hypolimnion, with highest concentrations in late summer and early fall (Fig. 5d).

Interestingly, when the lake became mixed again in November and December, it did not return to the oxic state but became anoxic throughout (Fig. 5b). Apparently, mixing during fall turnover was insufficient to bring the lake back into the oxic state; a behavior that looks remarkably similar to the hysteresis

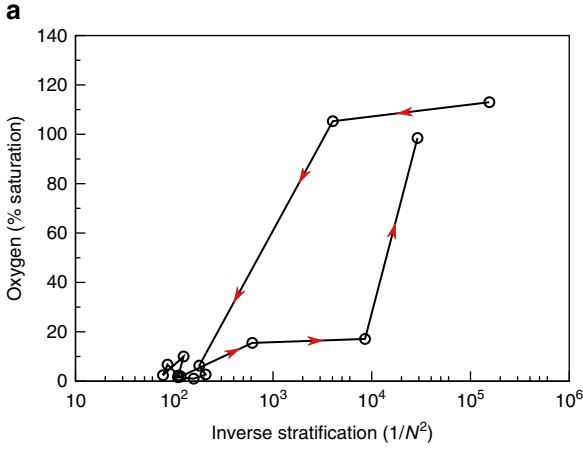

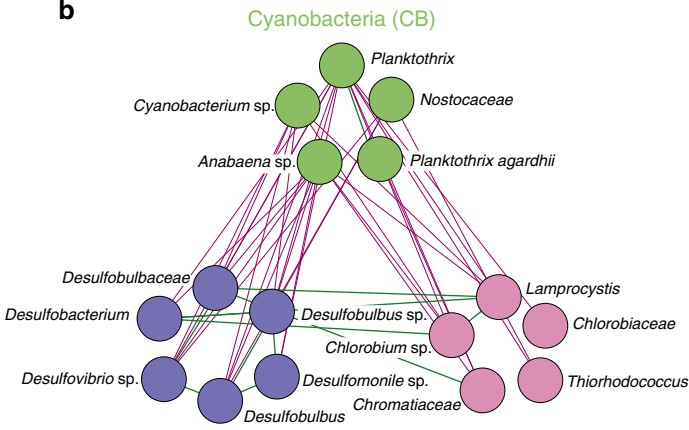

**Fig. 6** Evidence for regime shifts between oxic and anoxic states in Lake Vechten. **a** Hysteresis loop of oxygen saturation in the hypolimnion (7 m depth) plotted against the inverse of the stratification strength ($1/N^2$, where $N^2$ is the squared buoyancy frequency). The inverse of the stratification strength provides a simple proxy of oxygen diffusivity across the thermocline (see Methods). Data points are from March 2013 to March 2014; *red arrows* indicate the direction of time. **b** Co-occurrence network of bacteria in the metalimnion, based on 16S rRNA sequence data. *Green lines* represent positive interactions (co-occurrence), whereas *magenta lines* represent negative interactions (mutual exclusion)

predicted by our model. To study this pattern in further detail, we first calculated the squared buoyancy frequency, $N^2$, at the thermocline as a measure of the strength of stratification[29, 30]. The buoyancy frequency confirms that Lake Vechten is strongly stratified during the summer period, but well mixed during late fall and winter (Fig. 5e). The inverse of the squared buoyancy frequency ($1/N^2$) provides a simple proxy of the oxygen diffusivity across the thermocline (see Methods). Plotting the dissolved oxygen concentration in the hypolimnion against $1/N^2$ reveals a clear hysteresis loop (Fig. 6a), which is remarkably similar to the hysteresis loop predicted by the model (Fig. 3b). That is, intense mixing (high $1/N^2$) produced oxic conditions in Lake Vechten during the spring period, whereas the lake was anoxic at the same intensity of mixing during fall turnover. We note that this result is robust, irrespective of the exact depth at which the oxygen concentration is measured (Supplementary Fig. 1).

We used 16S rRNA amplicon sequencing data to assess whether changes in microbial community structure were consistent with the model predictions. Cyanobacteria dominated when the lake was well mixed and oxygenated in winter and early spring (Fig. 5f). Conversely, as the lake became stratified in summer, both sulfate-reducing bacteria and phototrophic sulfur bacteria increased in the anoxic hypolimnion whereas cyanobacteria decreased dramatically. Co-occurrence network analysis of microbial taxa in the metalimnion confirms this pattern: phototrophic sulfur bacteria coexisted with sulfate-reducing bacteria, while both groups were absent when cyanobacteria were present (Fig. 6b). Hence, the microbial community composition alternated between two distinct states, in agreement with the model predictions.

Interestingly, Lake Vechten did not turn anoxic during fall turnover every year[28]. In several earlier years, the entire water column became fully oxygenated in the fall and the sulfur bacteria disappeared. This matches our model predictions, which indicate that when a stratified lake with an oxic epilimnion and anoxic hypolimnion is mixed it may become either oxic or anoxic depending on the initial conditions (Fig. 2). That is, subtle differences in the mixing of these two water layers may determine whether the system develops towards an oxic or anoxic state. This bistable behavior is a typical feature of systems with alternative stable states.

## Discussion

Our model and lake data show that the interplay between microbial communities and oxidation-reduction processes creates systems with hysteresis loops and tipping points, in which gradual changes in oxygen influx, vertical stratification or nutrient levels can induce abrupt transitions between oxic and anoxic states. Other ecosystems with microbial communities similar to our model may undergo similar oxic-anoxic regime shifts. An interesting example is Lake Rogoznica, a marine lake along the Croatian coast filled with seawater[31, 32]. Similar to Lake Vechten, this enclosed marine ecosystem is stratified during summer, with an oxic epilimnion containing cyanobacteria and eukaryotic phytoplankton and an anoxic sulfidic hypolimnion dominated by phototrophic sulfur bacteria. In some years, but not all, the entire water column of Lake Rogoznica became anoxic during fall turnover[31], in agreement with the bistability predicted by our model. Furthermore, during anoxic holomixis, the phototrophic sulfur bacteria were replaced by sulfur-oxidizing chemotrophs[32], supporting one of the other model predictions, i.e., that environmental changes may alter the species composition of the sulfur bacteria in the anoxic state (Fig. 4).

In recent decades, hypoxia has spread across many eutrophied coastal waters[33, 34]. Prolonged hypoxia may develop into anoxic conditions with high sulfide concentrations, causing mass mortalities of fish and benthic organisms[33]. These coastal waters often show strong fluctuations in oxygen saturation, triggered by small changes in density stratification or coastal upwelling rates[35]. It is possible that the coastal waters in these highly dynamic regions are poised between the oxic and anoxic alternative stable states represented in our model. Our model results and field data support earlier suggestions[34, 36] that accumulation of sulfide and other reduced compounds and elimination of the aerobic community may produce a hysteresis loop, delaying recovery from coastal hypoxia.

We speculate that similar oxic-anoxic regime shifts may have occurred at a global scale in the Earth's geological past, when vast areas of the ocean became oxygen-depleted during oceanic anoxic events (OAEs)[37]. Many OAEs were associated with periods of global warming and high atmospheric $CO_2$ concentrations. High temperatures caused intense continental weathering, enhanced phosphorus discharge into the oceans, lowered oxygen solubility and increased thermohaline stratification suppressing

oxygen input into the deeper water layers[37, 38]. As our results indicate, these changes may trigger a regime shift to anoxic conditions. During OAEs, the oceans developed high sulfide concentrations[37, 39] and molecular biomarkers indicate that green sulfur bacteria were common in the photic zone[40–42]. These fascinating findings are all consistent with our model results, suggesting that OAEs are characterized by hysteresis effects and tipping points. If so, environmental changes that push the Earth's climate beyond a critical tipping point may cause large-scale transitions from oxic to anoxic conditions in the oceans that are not easily reversed.

Our model is only an abstract representation of the real world, providing a highly simplified picture of the complexity of oxic-anoxic transitions. We have specifically focused on microbially-mediated oxidation-reduction reactions in the sulfur cycle. Various other physical, chemical, and biological processes that may also affect whether ecosystems become oxic or anoxic have been conveniently ignored. Furthermore, the species composition of natural communities plays a key role not only in the sulfur cycle but also in several other biogeochemical cycles (e.g., in the nitrogen and carbon cycle), which may again lead to unexpected nonlinear feedback mechanisms. Hence, our findings may provide an interesting starting point for further integration of ecological community dynamics in biogeochemical process studies, and further analysis of its implications.

In conclusion, our model results and field data indicate that the well-known transition from oxic to anoxic conditions in aquatic environments is not a gradual process, but may occur in the form of a regime shift. This regime shift is mediated by nonlinear feedbacks between biogeochemical processes and microbial community dynamics, which can produce hysteresis. Once water becomes anoxic, a large oxygen influx is required before an aerobic community can become re-established, because the anaerobic sulfur cycle has to be overcome. Given the continued eutrophication of many lakes and coastal waters in combination with enhanced stratification by global warming, an improved understanding and prediction of oxic-anoxic regime shifts is essential if we are to mitigate the negative environmental effects of these phenomena.

## Methods

**Growth and inhibition functions**. The microbial literature offers several different equations for the growth and inhibition functions $g_j(X, Y)$ and $h_j(X)$. Here, we have chosen the Monod equation for growth, with multiplicative Monod kinetics when the growth rate is determined by two substrates[43]:

$$g_j(X, Y) = g_{max,j} \left( \frac{X}{K_{j,X} + X} \right) \left( \frac{Y}{K_{j,Y} + Y} \right) \tag{8}$$

where $g_{max,j}$ is the maximum specific growth rate of species $j$, and $K_{j,X}$ and $K_{j,Y}$ (μM) are the half-saturation constants of species $j$ on substrates $X$ and $Y$. Inhibition of microbial growth is described by the Haldane equation[44, 45]:

$$h_j(X) = \frac{1}{1 + (X/H_{j,X})} \tag{9}$$

where $H_{j,X}$ can be interpreted as a 'half-inhibition constant', i.e., it is the concentration of inhibitory substance $X$ at which the growth rate of species $j$ is reduced by 50%.

**Numerical simulation of the model**. Our model comprises 7 ordinary differential equations, each consisting of multiple nonlinear terms. We therefore relied on numerical analysis of the model behavior.

The 1D-bifurcation diagram in Fig. 3 was produced by numerical simulation of the dynamics until steady state, at different values of the oxygen diffusivity. We used a simple continuation approach to track the equilibria of the system, where the steady state of the previous simulation at a given oxygen diffusivity provided the initial conditions for the next simulation at a slightly higher (or slightly lower) oxygen diffusivity. In this way, the equilibrium of the sulfur bacteria was tracked by gradually increasing the oxygen diffusivity until the equilibrium of the sulfur

bacteria became unstable (i.e., until the trajectory diverged away from the equilibrium). Likewise, the alternative equilibrium of the cyanobacteria was tracked from the other side, by gradually decreasing the oxygen diffusivity, until the cyanobacterial equilibrium became unstable. This continuation approach was supplemented by additional numerical simulations sampling a broad range of initial conditions to verify the results. Trajectories always converged to a stable point; we did not observe stable periodic orbits or chaotic dynamics.

The 2D-bifurcation diagram in Fig. 4 was produced in a similar way. We first generated 1D-bifurcation diagrams as function of oxygen diffusivity, for a fixed value of the background phosphus (as in Fig. 3). This was repeated at many different values of the background phosphorus to produce the 2D-bifurcation diagram. In total, the graph in Fig. 4 is based on a grid of 400 × 400 numerical simulations.

All numerical simulations were run twice for consistency using two different methods of numerical integration: the integrate.odeint function from the widely used Python library Scipy, and a custom script in C for integrating systems of ordinary differential equations using the classic fourth order Runge–Kutta Method.

**Study site and sampling**. Lake Vechten (52°04′N, 5°05′E) has a maximum depth of 11.9 m, mean depth of 6.0 m and surface area of 4.7 ha[28]. The lake was sampled at biweekly to monthly intervals at every meter depth (0–10 m) from March 2013 to early April 2014. Water temperature and dissolved oxygen were measured on site using a Hydrolab DataSonde 4a (Hydrolab Corporation, Austin, TX, USA) and these data were visualized with Ocean Data View (version 4.6.5). Sulfate was measured by an auto-analyzer (SAN$^{++}$, Skalar, The Netherlands) based on the methylthymol blue method[46]. Sulfide was fixed with zinc acetate (10% w/v) immediately in the field, and subsequently measured spectroscopically in the laboratory using methylene blue[47]. Water samples were filtered through 0.2 μm nylon membrane filters (Millipore, GNWP) to collect bacterial cells. Filters were frozen immediately and stored at −20 °C until further processing.

**Buoyancy frequency**. The density of water, $\rho$ (kg m$^{-3}$), was calculated from temperature, $T$ (°C), using the Thiesen–Scheel–Diesselhorst equation[48]. We quantified the strength of stratification as the square of the buoyancy frequency[29, 30]:

$$N^2 = \frac{g}{\rho} \frac{d\rho}{dz} \tag{10}$$

where $z$ is depth (m), $g$ is the gravitational constant (9.8 m s$^{-2}$), and $d\rho/dz$ is the density gradient at the thermocline.

The flux of oxygen across the thermocline can be calculated as $F = K_z (\partial O_2/\partial z)$, where $K_z$ is the vertical eddy diffusivity. The eddy diffusivity depends on the buoyancy frequency according to $K_z = \Gamma \varepsilon/N^2$, where $\Gamma$ is the mixing efficiency and $\varepsilon$ is the rate of turbulent kinetic energy dissipation[30, 49]. Hence, the inverse of the squared buoyancy frequency ($1/N^2$) can be used as a simple proxy of the oxygen diffusivity across the thermocline.

**DNA extraction and 16S rRNA amplicon sequencing**. DNA was extracted from bacteria on the filters using the PowerSoil DNA Isolation Kit according to the manufacturer's instructions (Mo Bio, Laboratories Inc.). Extracted DNA concentrations were quantified with the Qubit dsDNA BR Assay Kit (Invitrogen). Sequencing was performed on an Illumina MiSeq system by the Research and Testing Laboratory (Lubbock, Texas, USA). The primer pair S-D-Bact-0341-b-S-17 (5′-CCTA CGGGNGGCWGCAG-3′) and S-D-Bact-0785-a-A-21 (5′-GACTACHVGGGTATCTAATCC-3′) was used to generate paired-end sequence reads covering the V3–V4 region of the 16S rRNA gene[50]. After quality filtering, a total of 2,934,111 sequences was obtained with an average sequence length of 420 bp.

**Co-occurrence network analysis**. 16S rRNA sequences were assigned to three functional groups: (1) Bacteria of the phylum *Cyanobacteria* (CB). (2) Phototrophic sulfur bacteria (PB) consisting of the phylum *Chlorobi* (green sulfur bacteria) and the order *Chromatiales* from the class *Gammaproteobacteria* (purple sulfur bacteria). (3) Sulfate-reducing bacteria (SB) consisting of the orders *Desulfobacterales*, *Desulfuromonadales*, and *Desulfovibrionales* from the class *Deltaproteobacteria*, as these were the only known sulfate-reducing bacteria present in the sequence data.

Co-occurrence networks were constructed with the Cytoscape plugin software program CoNet[51]. We used an ensemble approach to identify co-occurrence based on the Spearman rank correlation and Kullback–Leibler dissimilarity[51]. The ReBoot procedure with 4000 permutations was used to control for false-positive correlations. A false discovery rate of 5% was applied to control for multiple comparisons[52].

**Data availability**. The 16S amplicon sequences have been deposited in the Sequence Read Archive (SRA) of the National Center for Biotechnology Information (NCBI), as data set SAMN06314865-SAMN06314918. All other relevant data are available from the corresponding author on request.

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

## Acknowledgements

We thank Pieter Slot, Cherel Balkema, and Bas van Beusekom for assistance during fieldwork, Peter Serné for nutrient analysis, Karoline Faust for help with the

co-occurrence network analysis, André De Roos, Hal Caswell, Emily Melton, Elliot Marsden, Qian Li, and Andrew Free for useful discussions, and Petra Pjevac T.B., R.S., and G.M. were financially supported by the research priority area Systems Biology of the University of Amsterdam. In addition, T.B. was supported by the US Army Research Office under grant number 64052-MA and by HFSP under grant RGY0081/2012, M.D. was supported by a PhD scholarship from the China Scholarship Council (CSC), R.J.A. was supported by a Royal Society University Research Fellowship, and G.M. was supported by the ERC Advanced Grant PARASOL (No. 322551).

## Author contributions

G.M. and J.H. conceived the idea, and T.B. and R.J.A. added further suggestions. T.B., R.S., and J.H. designed the model. T.B. and J.H. performed the model simulations and analysis. M.D, R.S., and G.M. sampled the lake. M.D. performed the nutrient analysis. M.D. and G.M. performed the 16S rRNA sequence and network analysis. T.B., M.D., G.M., and J.H. wrote the manuscript, and all authors commented on the final version.

## Additional information

**Competing interests:** The authors declare no competing financial interests.

