## [Peer Review File · Nature Communications]

Reviewers' comments:

Reviewer #1 (Remarks to the Author):

This paper aims to explain rapid and robust changes in microbial communities composition and in the associated oxic/anoxic conditions in the water column (lakes, oceans). The novelty is mainly in the association of biogeochemical data, microbial data and a rather simple and general mathematical model for providing the underlying mechanism. This mechanism is based on the existence of a multi steady state in the model and the possibility to switch (and remain close) to one or the other steady state under changes of the oxic/anoxic conditions.

This underlying mechanism (and the model) is general enough to be used for several ecosystems at several scales and may be of interest for a large community. The discussion, including the consequences of the anoxia phenomenon which may result from global change, is dedicated to a large audience.

The demonstration in the manuscript that the model can explain the data (biogeochemical data and microbial community composition) is convincing, the choice of the simulations and as well as the choice of the data illustrations have been done adequately.

The underlying mechanism is based on the existence of an hysteresis phenomenon : once the oxygen has been depleted for some time, the system reaches the anoxic steady state. It needs a very large amount of oxygen i) to create an oxic steady state and ii) to push the system towards the attraction basin of this oxic state. The reverse is also true.

Many processes involving a shift regime seem to be based on this kind of mechanism, but to my knowledge, this paper convincingly suggests it for the first time for the anoxic/oxic changes, including the microbial communities.

Furthermore, the model includes the phosphorus dynamics and its effects on microbial communities, which allows us to anticipate the impact of eutrophication on oxic/anoxic shifts.

I found the paper clearly written and easily understandable.

As a consequence, I find that the manuscript deserves consideration for publication in a wide audience journal.

My following technical comments aim to improve the manuscript. I spent some time on numerical simulations to see the different steady states in the model and the hysteresis phenomenon and I realized that the list of parameters in table 1 and the captions of figure 2 are not sufficient to get figure 2. Could the author check the parameters values in figure 2 and give the exact values of the parameters which are only fixed through a range of values in Table 1? For instance, I took parameters like in Table 1 and captions in Figure 2, but I needed $N_{CB}=1e+11$ cells per liter (instead of $1e+09$ in the manuscript) to get qualitative results like in Figure 2 c-d.

My second comment is more on the comparison with real systems. The choice of numerical simulations made on Figure 2 is based on dynamics taking place on several years (at least 20 years for reaching the steady state in Figure 2 d for instance). Is this supported by observations in real systems? Actually, I think that a figure showing the different states of the system (using one state variable or something else) versus the oxygen diffusivity would be very illustrative.

The bifurcation diagram on Figure 4 is of bad quality, and this could be clearly improved. In general, algorithms are not provided (for the numerical simulations...). For instance, which methods are used to get the bifurcation diagram? This could be included in the SM.

Reviewer #2 (Remarks to the Author):

Review for 'Oxic-anoxic regime shifts induced by biogeochemical feedbacks in microbial communities'

In the submitted study, the authors introduce a relatively simple computational model to predict the state of the microbial community stratified freshwater systems under oxic and anoxic conditions. The model depends on biological parameters (growth, yield and inhibition) of such systems microbial key-players: cyanobacteria, sulfate-reducers and phototrophic sulfur oxidizers, and the availability of key substrates and nutrients: sulfur (reduced and oxidized), oxygen and phosphorus. With their model, the authors are able to show different microbial communities, referred to as alternative stable states, will establish in response to distinct changes in sulfur, oxygen and phosphorus availability. Further, they use oxygen and community composition data collected during an annual cycle in a small, seasonally stratified lake to demonstrate that microbial communities will behave similar to the models prediction.

I find the study appealing and valuable, in particular the aspect that this, or a similar derived model, could potentially be used to predict if experimental systems similar to Lake Vechten will to oxic or anoxic after, perturbation. However, I do have some concerns considering the presentation of the data and the interpretation of the presented data.

1. Title and abstract. Reading the title and the abstract the first time, I expected to read a study which clearly proves that changes in microbial community induce changes in the biogeochemical environment: 'Here, we present evidence that changes in microbial community composition can induce regime shifts in the biogeochemistry of aquatic ecosystems'. But the model and data presented to not prove this, in my opinion.

How I understand it, these claims are based on i.e. the following scenario presented on Page 7 and 8 of the manuscript: '... in the oxic state, the population density of cyanobacteria increases with phosphorus enrichment. High cyanobacterial densities can sustain the oxic state by their own photosynthetic oxygen production even when diffusive oxygen influx into the system becomes very low. Conversely, in the anoxic state, population densities of the anaerobic sulfur bacteria increase with phosphorus enrichment, and hence they can maintain anoxic conditions up to higher oxygen diffusivities'

However, a few sentences before, the authors themselves point out what is more correct then stating that microbial community composition induces regime shifts: 'Phosphorus enrichment has a profound effect on oxic-anoxic regime shifts.'

Similarly, they point out that environmental factors in deed do control community composition (and thus induce a regime shift) in other parts of the manuscript, e.g. 'Only when oxygen influx becomes high enough to suppress the phototrophic sulfur and sulfate-reducing bacteria, can the cyanobacteria invade'; 'Once water becomes anoxic, a large oxygen influx is required before an aerobic community can become re-established, because the anaerobic sulfur cycle has to be overcome'.

As I see it, it is not the microbial community that induces a regime shift. It is biogeochemical parameters (some of them modeled here) that are the determining factor to which alternative stable state (oxygenic phototrophs vs. anoxygenic sulfur bacteria) will prevail after a regime shift. The abstract (and parts of the introduction) should better reflect these findings, and maybe a more appropriate title could be selected.

(On a side note: while I see that the authors could argue against this, since oxygen and sulfur availability are strongly influenced by community composition itself, this is not the case for phosphorus)

2. Temperature as factor in the model, or explanatory variable for the lake data? In the discussion part of the manuscript, the authors give some great examples of how temperature affects a variety of ecosystems which can exist in alternative stable states, and put their research in the context of global change. This makes me wonder, should/can temperature be a variable in such a model?

Temperature does not only directly affect oxygen solvability, but also the rate of microbial processes. Thus I'm also wondering, could differences in water temperature (and their effect on process rates and thus the reduced sulfur pool, in particular) explain the observation '... intense mixing (high $1/N^2$) produced oxic conditions in Lake Vechten during the spring period, whereas the lake was anoxic at the same intensity of mixing during fall turnover.' If so (and based on literature and personal experience, I do think it is so), going oxic after spring mixing and anoxic after autumn mixing are not two alternative stable states that can arise from the same set of environmental conditions, as some highly influential environmental parameters (reduced sulfur pool, and temperature, which both are not reported) were different to begin with.

3. Could published data from other systems be plugged into the model to underline its value? I think it could greatly benefit the manuscript if authors could show that the model is correct on systems other than Lake Vechten. In particular because the authors' state '...that the results are quite robust, since we obtained qualitatively similar results when using parameter values representative of marine environments and microbial mats.'

4. Bring the key points into focus. After reading the manuscript several times, the 'take home message' I have is that while environmental factors are going to induce a regime shift and will determine which alternative stable (microbial community) state is going to arise, the composition of the microbial community is going to influence how stable this 'stable state' in deed is, or in other words – how much environmental perturbation is necessary to cause a reversal of the regime shift, or cause another regime shift. Is this the message the authors wanted to convey? I hope so, cause I like it a lot. However, it does not fit the title, nor is clear and easy to derive from (all of) the manuscripts text.

Kind regards,

Petra Pjevac

Reviewer #3 (Remarks to the Author):

This is an interesting manuscript that argues for specific microbial community feedbacks being established under anoxic conditions, which can be so strong that they may pervade under the breakdown of stratification, leading to more semi(permanent) anoxia. The evidence comes from a model that simulates the interactions between cyanobacteria, sulphate-reducing bacteria, and phototrophic sulphur bacteria, of which the latter two communities show evidence of hysteresis following the onset of anoxia. Empirical data, collected through DNA sequencing of microbial communities, shows that the patterns of dominance of the different microbial communities co-occur with the same oxic states the model predicts.

Overall the manuscript is well written and clearly structured. There are some issues that require further consideration and clarification though.

1. A crux of the manuscript seems to be the fact that Lake Veichten did not become oxygenated when it mixed again in November and December of the year of study. What about previous years? Was this year of study a one off? The authors consider the mixing potential through buoyancy measurements, and use these results to argue their model reflects reality. But – there are many reasons as to why a lake may not become oxygenated again. The microbe community is one reason but it would be useful if the authors could consider other limnological processes. Many lakes show enhanced anoxia through the summer period, eventually leading to permanent anoxia, despite mixing. This can be a slow or fast process, and is often associated with enhanced nutrient (P) recycling in the hypolimnion associated with anoxia – a positive feedback. There are many other limnological and biological processes that are impacted by this. So – the authors make a large leap when they argue that it could be solely due to the microbe communities. I suggest they explore other processes more, and/or refine their language in terms of explanatory power of their results.

2. Linked to point (1), I'd like to see more consideration given to the system drivers. Lakes typically respond to multiple slow and fast drivers, and simply increasing nutrients may make a lake tip, but there are many experiments that show it is the interaction of multiple drivers that lead to hysteresis. For example, this model is driven by P. What about N? Many lakes that are (becoming) eutrophic are actually N limited due to the relatively high abundance of P. What about other biological communities in the lake that could also be influencing the nutrient cycling and/or oxygen regimes?

3. Finally, although the model and empirical data do show interesting results which the authors argue impact on lake hysteresis, my view is that expanding this to the marine sphere and across deep time is a leap too far. I realise that everyone wants to speculate with their results and interpret them to make a big story, but I'd suggest they stick to what they know – lake systems. So I'd suggest they tone down the last part of the manuscript. No doubt microbial systems are important in these wider contexts, but the authors don't have the data to show that for these contexts, so it is purely speculation.

Response to Reviewer 1

“This paper aims to explain rapid and robust changes in microbial communities composition and in the associated oxic/anoxic conditions in the water column (lakes, oceans). The novelty is mainly in the association of biogeochemical data, microbial data and a rather simple and general mathematical model for providing the underlying mechanism. This mechanism is based on the existence of a multi steady state in the model and the possibility to switch (and remain close) to one or the other steady state under changes of the oxic/anoxic conditions.

This underlying mechanism (and the model) is general enough to be used for several ecosystems at several scales and may be of interest for a large community. The discussion, including the consequences of the anoxia phenomenon which may result from global change, is dedicated to a large audience.

The demonstration in the manuscript that the model can explain the data (biogeochemical data and

microbial community composition) is convincing, the choice of the simulations and as well as the choice of the data illustrations have been done adequately.

The underlying mechanism is based on the existence of an hysteresis phenomenon: once the oxygen has been depleted for some time, the system reaches the anoxic steady state. It needs a very large amount of oxygen i) to create an oxic steady state and ii) to push the system towards the attraction basin of this oxic state. The reverse is also true.

Many processes involving a shift regime seem to be based on this kind of mechanism, but to my knowledge, this paper convincingly suggests it for the first time for the anoxic/oxic changes, including the microbial communities.

Furthermore, the model includes the phosphorus dynamics and its effects on microbial communities, which allows us to anticipate the impact of eutrophication on oxic/anoxic shifts.

I found the paper clearly written and easily understandable.

As a consequence, I find that the manuscript deserves consideration for publication in a wide audience journal.”

We thank Reviewer 1 for his/her kind comments. Below, we outline our response to the helpful technical suggestions.

1. My following technical comments aim to improve the manuscript. I spent some time on numerical simulations to see the different steady states in the model and the hysteresis phenomenon and I realized that the list of parameters in table 1 and the captions of figure 2 are not sufficient to get figure 2. Could the author check the parameters values in figure 2 and give the exact values of the parameters which are only fixed through a range of values in Table 1? For instance, I took parameters like in Table 1 and captions in Figure 2, but I needed $N_{\text{CB}}=1e+11$ cells per liter (instead of $1e+09$ in the manuscript) to get qualitative results like in Figure 2 c-d.

Indeed, for some parameter values we gave only the ranges instead of the exact values in our previous submission. We thank Reviewer 1 for such careful attention to detail in noticing this error. We have now updated Table S1 and the figure captions to include the correct parameters to reproduce the exact quantitative results that are seen in Figure 2 and our other figures. We note that for two parameters (α_0 and P_b) we used a range of values in the bifurcation diagrams of Figures 3 and 4. For these two parameters we still provide the range of values in Table S1, but specify the exact values for the simulations of Figure 2 in the caption of Figure 2.

2. My second comment is more on the comparison with real systems. The choice of numerical simulations made on Figure 2 is based on dynamics taking place on several years (at least 20 years for reaching the steady state in Figure 2 d for instance). Is this supported by observations in real systems? Actually, I think that a figure showing the different states of the system (using one state variable or something else) versus the oxygen diffusivity would be very illustrative.

The slow dynamics were caused by the very low values for the diffusivities of sulfur and oxygen used in our model. In aquatic ecosystems, mixing of solutes is largely driven by turbulent diffusion (i.e., the motion caused by turbulent eddies) rather than by molecular diffusion. Turbulent diffusion in lakes and marine ecosystems can vary over several orders of magnitude, depending on wind and wave action, currents, convective heat flux, stratification, and so on. Our previous values were at the very low end of this broad range of variation. We have now increased the diffusivities of sulfur and oxygen by 2 orders of magnitude in Figure 2, which is still in the realistic range and speeds up the dynamics. The system now reaches steady state within a few months (instead of 20 years!), which indeed seems much more realistic for stratified lakes.

We changed not only the values of the diffusivities but also some of the other parameter values of the model to improve realism. This implied that we had to re-run all our simulations to make new versions of Figures 2, 3 and 4.

In Figure 3 we have plotted the different states of the system versus the oxygen diffusivity. This nicely illustrates the bifurcation pattern, a classic example of a catastrophe fold. At low diffusivity, the system becomes anoxic. At intermediate diffusivity, the system displays two alternative stable states. At high diffusivity, when the water column is intensely mixed, the dissolved oxygen concentration converges to the background oxygen concentration (i.e., the concentration in the water column equilibrates with the atmosphere).

3. The bifurcation diagram on Figure 4 is of bad quality, and this could be clearly improved. In general, algorithms are not provided (for the numerical simulations...). For instance, which methods are used to get the bifurcation diagram? This could be included in the SM.

We have included an updated 2D-bifurcation diagram at higher resolution and of better quality in Figure 4.

The 1D-bifurcation diagram in Figure 3 was produced by numerical simulation of the dynamics until steady state, at different values of the oxygen diffusivity. Basically, we used a simple continuation approach to track the equilibria of the system. That is, we used the steady state of the previous simulation at a given oxygen diffusivity as initial conditions for the next simulation at a slightly higher (or slightly lower) oxygen diffusivity. In this way, the equilibrium of the sulfur bacteria was tracked by gradually increasing the oxygen diffusivity until the equilibrium of the sulfur bacteria became unstable (i.e., until the trajectory diverged away from the equilibrium). We note that our system consists of 7 ODEs, and hence stability of point equilibria was assessed numerically by divergence or convergence of the trajectories. Likewise, the alternative equilibrium of the cyanobacteria was tracked from the other side, by gradually decreasing the oxygen diffusivity, until the cyanobacterial equilibrium became unstable. This continuation approach was supplemented by additional numerical simulations sampling a broad range of initial conditions to verify the results. Trajectories always converged to a stable point; we did not observe any stable periodic orbits or chaotic dynamics.

The 2D-bifurcation diagram in Figure 4 was produced in a similar way. We first generated 1D-bifurcation diagrams as function of oxygen diffusivity, for a fixed value of the background phosphorus (as in Figure 3). This was repeated at many different values of the background phosphorus to produce the 2D-bifurcation diagram. In total, the graph in Figure 4 is based on a grid of 400 x 400 numerical simulations.

*All numerical simulations were produced twice for consistency using two different methods of numerical integration: 1) the `integrate.odeint` function from the widely used Python library `Scipy` and 2) a custom script in C for integrating systems of ordinary differential equations using the classic 4th order Runge-Kutta Method. We have now included a description of these numerical methods in the *Supplementary Information*.*

Response to Reviewer 2

In the submitted study, the authors introduce a relatively simple computational model to predict the state of the microbial community stratified freshwater systems under oxic and anoxic conditions. The model depends on biological parameters (growth, yield and inhibition) of such systems microbial key-players: cyanobacteria, sulfate-reducers and phototrophic sulfur oxidizers, and the availability of key substrates and nutrients: sulfur (reduced and oxidized), oxygen and phosphorus. With their model, the authors are able to show different microbial communities, referred to as alternative stable states, will establish in response to distinct changes in sulfur, oxygen and phosphorus availability. Further, they use oxygen and community composition data collected during an annual cycle in a small, seasonally stratified lake to demonstrate that microbial communities will behave similar to the models prediction.

I find the study appealing and valuable, in particular the aspect that this, or a similar derived model, could potentially be used to predict if experimental systems similar to Lake Vechten will to oxic or anoxic after, perturbation. However, I do have some concerns considering the presentation of the data and the interpretation of the presented data.

We thank Reviewer 2 for the kind words and constructive comments.

1. Title and abstract. Reading the title and the abstract the first time, I expected to read a study which clearly proves that changes in microbial community induce changes in the biogeochemical environment: ‘Here, we present evidence that changes in microbial community composition can induce regime shifts in the biogeochemistry of aquatic ecosystems’. But the model and data presented to not prove this, in my opinion. How I understand it, these claims are based on i.e. the following scenario presented on Page 7 and 8 of the manuscript: ‘... in the oxic state, the population density of cyanobacteria increases with phosphorus enrichment. High cyanobacterial densities can sustain the oxic state by their own photosynthetic oxygen production even when diffusive oxygen influx into the system becomes very low. Conversely, in the anoxic state, population densities of the anaerobic sulfur bacteria increase with phosphorus enrichment, and hence they can maintain anoxic conditions up to higher oxygen diffusivities’

However, a few sentences before, the authors themselves point out what is more correct then stating that microbial community composition induces regime shifts: ‘Phosphorus enrichment has a profound effect on oxic-anoxic regime shifts.’”

Reviewer 2 is quite correct that our wording was somewhat imprecise. Oxic-anoxic regime shifts in our model are driven by environmental changes, which modify the relative contribution of the oxygen-producing and oxygen-consuming processes in the system. More specifically, in our model the feedbacks between biogeochemical processes and microbial community dynamics lead to a system with alternative stable states. That is, if cyanobacteria are dominant first, they can produce sufficient oxygen to suppress the sulfur bacteria and the system remains oxic. Conversely, if the sulfur bacteria are dominant first, they can produce sufficient sulfide to suppress the cyanobacteria, and the system remains anoxic. Changes in environmental parameters (like oxygen diffusivity, phosphorus load, etc) modify the relative contributions of these different biogeochemical processes and microbial populations, and can thereby push the system from the oxic to the anoxic state or vice versa. We did attempt to describe this point carefully in the main text, as indicated by the above quote from page 7 and 8 of the manuscript.

We agree that our previous wording in the Abstract and title were ambiguous, as pointed out by the reviewer. Therefore, we have changed the title of our manuscript to: “Oxic-anoxic regime shifts mediated by feedbacks between biogeochemical processes and microbial community dynamics”.

Furthermore, we rewrote the sentence in the Abstract indicated by the reviewer. This sentence now reads: “Here, we present evidence that gradual environmental changes (e.g., enhanced stratification, nutrient enrichment) can induce major regime shifts between oxic and anoxic states, mediated by the interplay between microbial community dynamics and biogeochemical oxidation-reduction processes.”

We also changed some of the wording in the main text to improve the clarity of our manuscript. For instance, we now provide a better explanation of the alternative stable states predicted by the model (page 6).

2. Temperature as factor in the model, or explanatory variable for the lake data? In the discussion part of the manuscript, the authors give some great examples of how temperature affects a variety of ecosystems which can exist in alternative stable states, and put their research in the context of global change. This makes me wonder, should/can temperature be a variable in such a model?

Temperature does not only directly affect oxygen solvability, but also the rate of microbial processes. Thus I’m also wondering, could differences in water temperature (and their effect on process rates and thus the reduced sulfur pool, in particular) explain the observation ‘... intense mixing (high $1/N^2$) produced oxic conditions in Lake Vechten during the spring period, whereas the lake was anoxic at the same intensity of mixing during fall turnover.’ If so (and based on literature and personal experience, I do think it is so), going oxic after spring mixing and anoxic after autumn mixing are not two alternative stable states that can arise from the same set of environmental conditions, as some highly influential environmental parameters (reduced sulfur pool, and temperature, which both are not reported) were different to begin with.”

We appreciate the suggestion to include the effects of temperature on oxygen solubility and on the rates of microbial processes in the model. Such a change in model structure is beyond the scope of the present paper, however, because temperature affects not only oxygen solubility and microbial kinetics but also other important aspects such as the stratification of aquatic ecosystems. All mathematical models are abstractions of reality that emphasize some aspects while simplifying or ignoring other environmental processes. In this case we believe that the focus of our paper should be the regime shifts generated by the interaction between biogeochemical processes and microbial community dynamics. We are planning to produce a follow-up study investigating these regime shifts in a more complex 1D vertical model (based on the Navier-Stokes equations), where we want to include the vertical temperature gradient.

We have now added a new paragraph, on page 11, in which we explain that our model is only an abstract representation of the real world, providing a highly simplified picture of the complexity of oxic-anoxic transitions. We have specifically focused on microbially-mediated oxidation-reduction reactions in the sulfur cycle, while ignoring other processes that may also affect whether systems become oxic or anoxic. However, we believe that the overall consistency of our model predictions with the mixing data, oxygen data, sulfur data and microbial community data provides considerable support for an oxic-anoxic regime shift.

We note that lake data of temperature, reduced sulfur (sulfide) and oxidized sulfur (sulfate) are reported in our manuscript. They are presented in Figures 5a, 5c and 5d, and described on page 8. The data show that water temperature during spring turnover (March) and fall turnover (late November) was quite similar (Figure 5a). It was perhaps slightly colder in spring, but this minor temperature difference is unlikely to explain the large difference in dissolved oxygen concentrations.

“3. Could published data from other systems be plugged into the model to underline its value? I think it could greatly benefit the manuscript if authors could show that the model is correct on systems other than Lake Vechten. In particular because the authors’ state ‘...that the results are quite robust, since we obtained qualitatively similar results when using parameter values representative of marine environments and microbial mats.’”

Detailed time series data of microbial community dynamics during oxic-anoxic transitions are rare. However, an interesting example is provided by Lake Rogoznica, a marine lake along the Croatian coast filled with seawater (e.g., Pjevac et al. 2015). Similar to Lake Vechten, this enclosed marine ecosystem is stratified during summer, with an oxic epilimnion containing cyanobacteria and eukaryotic phytoplankton and an anoxic sulfidic hypolimnion dominated by phototrophic sulfur bacteria. In several years, but not all, the entire water column of Lake Rogoznica became anoxic during fall turnover, in agreement with the bistability predicted by our model. During these periods of anoxic holomixis, phototrophic sulfur bacteria were replaced by sulfur-oxidizing chemotrophs (Pjevac et al 2015), supporting the model prediction that environmental changes may alter the species composition of the sulfur bacteria in the anoxic state (Figure 4). We have now highlighted the interesting data from this marine lake in the section “Oxic-anoxic regime shifts in other ecosystems” (page 10).

“4. Bring the key points into focus. After reading the manuscript several times, the ‘take home message’ I have is that while environmental factors are going to induce a regime shift and will determine which alternative stable (microbial community) state is going to arise, the composition of the microbial community is going to influence how stable this ‘stable state’ in deed is, or in other words – how much environmental perturbation is necessary to cause a reversal of the regime shift, or cause another regime shift. Is this the message the authors wanted to convey? I hope so, cause I like it a lot. However, it does not fit the title, nor is clear and easy to derive from (all of) the manuscripts text.”

Reviewer 2 is largely correct in her interpretation of our 'take home message' and we are very pleased that she likes it so much. Our results show that gradual environmental changes (e.g., enhanced stratification, nutrient enrichment) can induce major regime shifts between oxic and anoxic states, mediated by the interplay between microbial community dynamics and biogeochemical oxidation-reduction processes. As mentioned before, we agree that our previous title did not reflect this take-home message very well and have changed the title of the manuscript and wording in the Abstract to better reflect this.

One subtle point that is somewhat different from the reviewer's description is that the existence of alternative stable states implies (by definition) that the initial conditions will determine which of these alternative stable states will be reached. In the region with alternative stable states, our model predicts that if sulfate-reducing and phototrophic sulfur bacteria are initially dominant, the concentration of reduced sulfur becomes sufficiently high to suppress cyanobacterial growth, and the system develops an anoxic state (Fig. 2a,b). Conversely, if cyanobacteria are dominant first, their photosynthetic oxygen production is sufficiently high to suppress the growth of sulfate-reducing bacteria and phototrophic sulfur bacteria, generating an oxic state (Fig. 2c,d). We have changed the text on page 6 accordingly to clarify this point.

Reviewer 3 Comments

“This is an interesting manuscript that argues for specific microbial community feedbacks being established under anoxic conditions, which can be so strong that they may pervade under the breakdown of stratification, leading to more semi(permanent) anoxia. The evidence comes from a model that simulates the interactions between cyanobacteria, sulphate-reducing bacteria, and phototrophic sulphur bacteria, of which the latter two communities show evidence of hysteresis following the onset of anoxia. Empirical data, collected through DNA sequencing of microbial communities, shows that the patterns of dominance of the different microbial communities co-occur with the same oxic states the model predicts.

Overall the manuscript is well written and clearly structured. There are some issues that require further consideration and clarification though.

We thank Reviewer 3 for these kind words. Below, we outline our response to his/her helpful suggestions.

1. A crux of the manuscript seems to be the fact that Lake Vechten did not become oxygenated when it mixed again in November and December of the year of study. What about previous years? Was this year of study a one off? The authors consider the mixing potential through buoyancy measurements, and use these results to argue their model reflects reality. But – there are many reasons as to why a lake may not become oxygenated again. The microbe community is one reason but it would be useful if the authors could consider other limnological processes. Many lakes show enhanced anoxia through the summer period, eventually leading to permanent anoxia, despite mixing. This can be a slow or fast process, and is often associated with enhanced nutrient (P) recycling in the hypolimnion associated with anoxia – a positive feedback. There are many other limnological and biological processes that are impacted by this. So – the authors make a large leap when they argue that it could be solely due to the microbe communities. I suggest they explore other processes more, and/or refine their language in terms of explanatory power of their results.

Good question! Lake Vechten did not turn anoxic during fall turnover every year (e.g., Steenbergen & Verdouw 1982). In several earlier years, the entire water column became fully oxygenated in the fall and the sulfur bacteria disappeared. This matches our model predictions, which indicate that when a stratified lake with an oxic epilimnion and anoxic hypolimnion is mixed it may become either oxic or anoxic depending on the initial conditions (Fig. 2). That is, subtle differences in the mixing of these two water layers may determine whether the system develops towards an oxic or anoxic state. We have now added a paragraph to explain that such bistable behaviour is a typical feature of systems with alternative stable states (page 9).

The same phenomenon is known from other lakes, such as Lake Rogoznica which also becomes anoxic during fall turnover in some years but not in all years. See page 10.

We fully agree with the reviewer that there are a variety of other physical, chemical and biological processes that may also determine whether a lake becomes anoxic. In fact, also in our model it is not solely the microbial community but the interaction of the microbial community dynamics with biogeochemical

processes (as summarized in Figure 1) that produces a system with oxic/anoxic regime shifts. We have now adapted the title of our manuscript and several sections of the main text to refine our language and improve the clarity of our explanations (see also our responses to reviewers #1 and #2).

“2. Linked to point (1), I’d like to see more consideration given to the system drivers. Lakes typically respond to multiple slow and fast drivers, and simply increasing nutrients may make a lake tip, but there are many experiments that show it is the interaction of multiple drivers that lead to hysteresis. For example, this model is driven by P. What about N? Many lakes that are (becoming) eutrophic are actually N limited due to the relatively high abundance of P. What about other biological communities in the lake that could also be influencing the nutrient cycling and/or oxygen regimes?”

The function of P in the model is primarily to represent a limiting nutrient. If the model was abundant in P but limited by another key nutrient (such as N), in principle it would produce similar regime shifts. For other reasons, however, the use of N as limiting nutrient is a bit more complicated. In contrast to P, the nitrogen cycle involves a series of oxidation-reduction processes mediated by microorganisms (nitrification, denitrification, nitrogen fixation, and so on). Therefore, combining the oxidation-reduction reactions in the sulfur cycle with those in the nitrogen cycle would lead to a more complex model. We now briefly mention that the involvement of microbial communities in the N cycle may again lead to interesting feedback mechanisms (page 11).

We fully agree with the reviewer that other biological communities may also influence nutrient cycling and/or oxygen regimes in real systems. The reason we selected photosynthetic sulfur bacteria, sulfate-reducing bacteria and cyanobacteria for our model is that they are key to the sulfur cycle and photosynthetic oxygen production in aquatic ecosystems, which was the focus of our study. We are fully aware that our model, like all models, is a simplification of reality and that lakes may respond to multiple fast and slow drivers affecting oxygen levels. Therefore, we have added a new paragraph where we explicitly mention that our model is only a simplification of reality and that several other physical, chemical and biological processes may also affect whether a system becomes oxic or anoxic (page 11).

“3. Finally, although the model and empirical data do show interesting results which the authors argue impact on lake hysteresis, my view is that expanding this to the marine sphere and across deep time is a leap too far. I realise that everyone wants to speculate with their results and interpret them to make a big story, but I’d suggest they stick to what they know – lake systems. So I’d suggest they tone down the last part of the manuscript. No doubt microbial systems are important in these wider contexts, but the authors don’t have the data to show that for these contexts, so it is purely speculation.”

We do agree with Reviewer 3 to some extent, but not completely. Most importantly, the mechanisms incorporated in our model are not specific for freshwater lakes. Cyanobacteria, photosynthetic sulfur bacteria and sulfate-reducing bacteria are also common in marine ecosystems, sulfur is one of the most abundant elements in seawater (after sodium and chloride), and marine ecosystems can also become anoxic and sulfidic. Furthermore, we note that Reviewer 1 and Reviewer 2 were enthusiastic about the comparison of our model predictions with marine ecosystems.

We have now shown that our model predictions can be extended to the marine sphere by inclusion of the marine lake Rogoznica in response to the helpful comments of Reviewer 2. Although there are some differences, the overall dynamics of this marine lake are quite similar to our observations in Lake Vechten (page 10, top paragraph).

In retrospect, we felt that our extrapolation to the Baltic Sea deepwater oxygenation (BOX) project was not sufficiently convincing. This system deviates from natural systems, because it was artificially oxygenated. We could not demonstrate hysteresis for the BOX project, and hence our discussion of this example from the Baltic Sea may indeed have been a leap too far. Hence, in line with the advice of Reviewer #3, this example has been removed.

We believe that it would be awkward if a manuscript on oxic-anoxic regime shifts involving the sulfur cycle would not mention the spread of hypoxia across many eutrophied coastal waters, where high sulfide concentrations cause mass mortalities of fish and benthic organisms. Furthermore, earlier studies have suggested that hypoxia in coastal waters may also display hysteresis effects (Middelburg & Levin 2009; Zhang et al. 2010), and have sketched a verbal explanation of the interplay between biological activity and

biogeochemical processes that is quite akin to our model predictions. Therefore, we feel that our manuscript would be incomplete if we would not cite these earlier studies of coastal hypoxia (page 10, second paragraph).

We agree that extrapolation of our results across deep time to oceanic anoxic events is speculative, although there are many interesting parallels that appear consistent with our model results. Hence, we now start this paragraph by the words “We speculate”, to clarify to the readers that this is an interesting but more speculative paragraph (page 10, third paragraph). Furthermore, we have shortened this paragraph by removing the last part on the mass extinctions caused by oceanic anoxic events, because we agree that this reference to the Permian mass extinction was indeed over the top.

Hence, we have toned down those sections in the last part of the manuscript where we felt our examples were not convincing or indeed a leap too far. However, we have retained and strengthened those sections that could be linked to our model results.

Reviewers' Comments:

Reviewer #1:

Remarks to the Author:

I want to thank the author for having adequately answered to my comments on the previous version, the new figures are much more adapted to the NC journal. The technical details for the Supplementary material allow to redo the simulations with comparable tools, it is also interesting to notice that the simulations have been made using several schemes to check consistency. The new version of the paper has been improved and is more focused than the previous one. I think that it is a very interesting contribution, deserving publication in the NC journal.

Reviewer #2:

Remarks to the Author:

The Authors have sufficiently addressed my concerns considering the wording of the abstract and certain passages of the text. The clarity of the entire text is much improved in comparison to the primary manuscript version.

I'm also in favor of the new suggested title, it fits the described modeled and observed scenarios better.

I get that including temperature as a parameter in the model would in essence mean the whole study is redone from scratch. I understand this is not feasible. However, I'd like to highlight that the rerun of the model based on the remarks from reviewer 1 has resulted in a real nice improvement, aligning the model so much better to environmental data, and I applaud the effort the authors invested here!

While I disagree with reviewer 3 about 'extending the model on marine systems and across depths' being a leap to far, I think the authors have found a good balance and still manage to show how similar processes, albeit not the exact same mechanisms described in this model, cause extended anoxia and hypoxia in the marine environment. Well done!

Reviewer #4:

Remarks to the Author:

This paper presents an interesting and intricate interaction of 3 elemental cycles and 3 functional groups of bacteria. The analysis shows that alternate states of oxic / anoxic conditions can emerge under realistic conditions, and corroborates this assertion with field data from lake as well as discussion of other field situations. The paper is clearly written and easy to follow. Apparently the paper has already been through review by three referees who had positive opinions. I am a fourth review, added in the second round.

In my opinion the paper is technically sound and will be interesting to a wide readership.

My main suggestion is that the model description be compressed to text plus a fully-worded and more informative version of Figure 1. The equations can be moved to the online material. Readers of this manuscript will want to move quickly into the story. At the level of presentation in the text, the equations are almost useless. The growth and inhibition functions are not even written out, and these are the key to understanding stability. In a short-format journal like N Comms it will be impossible to explain the model in sufficient detail in main text. Everyone who has a keen interest in the model details will read the SI anyway. Best to make the main text as readable as possible for a general audience by moving the equations out.

Rebuttal to the reviewer's comments

Reviewer #1 (Remarks to the Author):

I want to thank the author for having adequately answered to my comments on the previous version, the new figures are much more adapted to the NC journal. The technical details for the Supplementary material allow to redo the simulations with comparable tools, it is also interesting to notice that the simulations have been made using several schemes to check consistency. The new version of the paper has been improved and is more focused than the previous one. I think that it is a very interesting contribution, deserving publication in the NC journal.

Rebuttal: We thank reviewer #1 for his kind remarks and are glad to read that he thinks that it is a very interesting contribution, which deserves publication.

Reviewer #2 (Remarks to the Author):

The authors have sufficiently addressed my concerns considering the wording of the abstract and certain passages of the text. The clarity of the entire text is much improved in comparison to the primary manuscript version. I'm also in favor of the new suggested title; it fits the described modeled and observed scenarios better.

I get that including temperature as a parameter in the model would in essence mean the whole study is redone from scratch. I understand this is not feasible. However, I'd like to highlight that the rerun of the model based on the remarks from reviewer 1 has resulted in a real nice improvement, aligning the model so much better to environmental data, and I applaud the effort the authors invested here!

While I disagree with reviewer 3 about 'extending the model on marine systems and across depths' being a leap to far, I think the authors have found a good balance and still manage to show how similar processes, albeit not the exact same mechanisms described in this model, cause extended anoxia and hypoxia in the marine environment. Well done!

Rebuttal: We are glad to read that we addressed all the concerns of reviewer #2 and that the manuscript improved drastically.

Reviewer #4 (Remarks to the Author):

This paper presents an interesting and intricate interaction of 3 elemental cycles and 3 functional groups of bacteria. The analysis shows that alternate states of oxic / anoxic conditions can emerge under realistic conditions, and corroborates this assertion with field data from lake as well as discussion of other field situations. The paper is clearly written and easy to follow. Apparently the paper has already been through review by three referees who had positive opinions. I am a fourth review, added in the second round.

In my opinion the paper is technically sound and will be interesting to a wide readership.

My main suggestion is that the model description be compressed to text plus a fully-worded and more informative version of Figure 1. The equations can be moved to the online material. Readers of this manuscript will want to move quickly into the story. At the level of presentation in the text, the equations are almost useless. The growth and inhibition functions are not even written out,

and these are the key to understanding stability. In a short-format journal like N Comms it will be impossible to explain the model in sufficient detail in main text. Everyone who has a keen interest in the model details will read the SI anyway. Best to make the main text as readable as possible for a general audience by moving the equations out.

Rebuttal: We thank reviewer #4 for his or her kind remarks and his opinion that the paper will be interesting for a wide readership. His or her suggestion to move the equations to the online material was overruled by the editor. We have added a sentence in the main text, below equations (1)-(3), which explains that the growth and inhibition functions are detailed in the Methods section.